# Wildfire Smoke Adjustment Factors for Low-Cost and Professional PM_2.5_ Monitors with Optical Sensors

**DOI:** 10.3390/s20133683

**Published:** 2020-06-30

**Authors:** William W. Delp, Brett C. Singer

**Affiliations:** Indoor Environment Group and Residential Building Systems Group, Lawrence Berkeley National Laboratory, Berkeley, CA 94720, USA; wwdelp@lbl.gov

**Keywords:** fine particles, air pollutant exposure, air quality monitoring, climate change impacts, health hazard assessment, respiratory health

## Abstract

Air quality monitors using low-cost optical PM_2.5_ sensors can track the dispersion of wildfire smoke; but quantitative hazard assessment requires a smoke-specific adjustment factor (AF). This study determined AFs for three professional-grade devices and four monitors with low-cost sensors based on measurements inside a well-ventilated lab impacted by the 2018 Camp Fire in California (USA). Using the Thermo TEOM-FDMS as reference, AFs of professional monitors were 0.85 for Grimm mini wide-range aerosol spectrometer, 0.25 for TSI DustTrak, and 0.53 for Thermo pDR1500; AFs for low-cost monitors were 0.59 for AirVisual Pro, 0.48 for PurpleAir Indoor, 0.46 for Air Quality Egg, and 0.60 for eLichens Indoor Air Quality Pro Station. We also compared public data from 53 PurpleAir PA-II monitors to 12 nearby regulatory monitoring stations impacted by Camp Fire smoke and devices near stations impacted by the Carr and Mendocino Complex Fires in California and the Pole Creek Fire in Utah. Camp Fire AFs varied by day and location, with median (interquartile) of 0.48 (0.44–0.53). Adjusted PA-II 4-h average data were generally within ±20% of PM_2.5_ reported by the monitoring stations. Adjustment improved the accuracy of Air Quality Index (AQI) hazard level reporting, e.g., from 14% to 84% correct in Sacramento during the Camp Fire.

## 1. Introduction

Throughout the Western U.S., wildland fires have increased in frequency and intensity over the past several decades due to climate change and the legacy of forest fire suppression [1,2,3,4,5]. Development at the wildland urban interface also has contributed to wildfire frequency and increased their cost in terms of human life and health and property damage [4,6].

Wildfire smoke contains fine particulate matter (PM_2.5_), toxic particle-phase constituents, ultrafine particles, and many irritant gases including acrolein and formaldehyde. Exposure to elevated levels of wildfire smoke specifically has been linked to many adverse health outcomes [7,8,9,10,11]. During wildfire smoke episodes, air pollutant concentrations can increase substantially, and low-income homes, which typically have high rates of uncontrolled air leakage, are particularly vulnerable [12]. Filtration can be cost-effectively applied to reduce exposures and health impacts of wildfire smoke in buildings [13,14].

PM_2.5_ is often used to track wildfire smoke because it is an established health hazard and is routinely measured at regulatory air quality monitoring stations. In the U.S., PM_2.5_ is regulated under the Clean Air Act, with health-based standards of 12 µg m^−3^ annual average and 35 µg m^−3^ averaged over 24 h [15]. The thresholds are set by the US EPA based on a systematic review of studies that examine how deaths, strokes, and other indicators such as hospitalizations increase as ambient PM2.5 increases [16]. To provide information to the public about the hazard posed by PM2.5, both below and above the thresholds of the standard, EPA uses the air quality index, or AQI [17]. The AQI is a piecemeal linear scale that relates PM2.5 concentrations to hazard level for sensitive subgroups and the general population. The AQI is normally calculated on a daily basis; but the interval can be shortened to as little as 3 h when ambient conditions change rapidly. It is also important to note that despite the cautions communicated about the AQI, wildfire smoke may still present substantial health risk and cause harm to individuals with pre-existing health conditions and vulnerabilities beyond that presented in the PM2.5 AQI.

The U.S. Federal Reference Method (FRM) for determining fine particle concentrations is gravimetric: it requires use of certified equipment to collect particles onto a filter which is equilibrated to standard temperature and humidity conditions and weighed before and after air sampling [18,19]. Devices that use alternative measurement methods—typically with the goal of achieving hourly or more resolved data—can be approved as Federal Equivalent Methods (FEM) by producing similar results when collocated with FRM sampling at multiple sites across multiple seasons. Agreement is acceptable when linear correlation is high (r > 0.97) with a slope of 1.00 ± 0.05 and intercept ≤ ±1 μg/m^3^. In practice, FEM monitors don’t always match collocated FRM measurements to these specifications [20]. Also, since FEM certification is based on seasonal statistics of daily averages, accuracy and precision may be lower for shorter averaging times.

Historically, it has been difficult to map the hazards posed by wildfire smoke because regulatory air monitoring stations are sparse in many areas. During severe events, portable monitors have been deployed to fill critical gaps in spatial coverage [21,22]; but that approach has been limited by the high cost of purchasing and maintaining equipment that meet data quality standards [23]. At least one low-cost monitor has been developed for the stated objective of filling this gap for wildfire smoke monitoring [24]; but the extent of its use to date is unclear. One study used surface measurements from regulatory air monitoring stations and low-cost particle monitor networks to translate satellite images of aerosol optical depth into PM_2.5_ for mapping of wildfire spread [25]. In that study, PurpleAir II monitors were compared to co-located FEM data but the evaluation did not include high PM_2.5_ concentrations (>100 µg/m^3^) or PM known to be predominantly wood smoke.

For years, industrial hygienists and researchers have measured PM_2.5_ inside and outside buildings using real-time monitors that quantify aerosol concentrations by light scattering. These devices are robust, have a wide dynamic range (3 or more orders of magnitude), and temporal resolution on the order of seconds. However, the intensity of scattered light depends on aerosol properties including size, shape, density, and refractive index [26,27]. Photometers are a main variant of optical monitor; they measure and translate scattering from ensembles of particles to an estimated mass concentration based on calibration with a defined aerosol [28,29]. Optical particle counters interpret scattering generated from single particles moving through a laser to estimate particle size based on assumptions about shape and optical properties; counts are aggregated over time to quantify number density by size. Aerosol mass concentrations are estimated based on assumptions or collocated measurements to determine particle density. Professional grade air quality monitors provide active flow control, sheath airflow, and component diagnostic monitoring to achieve high consistency and durability; but they cost thousands of dollars per unit and still require source-specific calibration factors or coincident gravimetric sampling.

Collections of low-cost air quality monitors that are being deployed throughout the world [30,31,32,33] could be used to track wildfire smoke events. However, the optical sensors used in the monitors vary with aerosol properties [34,35,36,37,38,39] and adjustment factors specific to wildfire smoke are needed.

The present study had the following objectives: (1) using data collected during the November 2018 Camp Fire in Northern California, determine adjustment factors (AFs) for four low-cost and three professional air quality monitors to improve their accuracy for measuring infiltrating PM_2.5_ associated with wildfire smoke; (2) using publicly available data from regulatory air quality monitoring stations (AQS) and nearby PurpleAir PA-II monitors at varied distances downwind from the fire, evaluate the variability of AFs for this monitor across space and time for sites throughout the region; (3) quantify PA-II AFs for smoke from two other recent wildfires and compare them to AFs determined for the Camp Fire; and (4) quantify the improvement in exposure estimates and AQI scores when the regional AF is applied to individual monitors throughout the region.

## 2. Methods

### 2.1. Overview

During the Camp Fire in November 2018, air quality monitors were operated at Lawrence Berkeley National Lab (LBNL) inside a small building with high infiltration air exchange and the doors to outdoors propped open at times to promote infiltration of outdoor air and particulate matter (PM). The monitors included one certified as a US EPA FEM, a wide range aerosol spectrometer designed for indoor air quality research, two professional grade photometers, and four low-cost monitors that use mass produced optical sensors. The latter group included three monitors purchased via retail distribution in the U.S. and production units of an IAQ station for networked building monitoring provided by the device maker (eLichens). PM_2.5_ concentrations reported by the alternative monitors were compared to those of the FEM to quantify response and adjustment factors. The monitors are listed in Table 1.

Additionally, we extracted and analyzed publicly available data collected at regulatory air quality monitoring stations (AQS) and nearby PurpleAir PA-II monitors for three Western US wildfires. For the Camp Fire, data were obtained for 12 Northern California (NorCal) AQS sites with high PM_2.5_ concentrations and 53 PA-II monitors near the 12 sites. Data were identified and analyzed for a single AQS site and nearby PA-II monitor that were in an area impacted by the Carr and Mendocino Complex Fires in California and for a single AQS and PA-II monitor impacted by the Pole Creek Fire in Utah.

### 2.2. Wildland Fires

#### 2.2.1. Camp Fire

At roughly 6:30 a.m. on November 8, 2018 a wildland fire started along Camp Creek Road near Poe Dam in Butte County, Northern California. The fire spread quickly and ravaged the nearby town of Paradise. It caused at least 86 fatalities and destroyed almost 19,000 buildings, many in the first few hours. It ultimately burned over 150,000 acres and was not contained until November 25. A map showing the area burned in the fire is provided in Figure 1.

Strong off-shore katabatic winds on November 8 carried smoke throughout the most heavily populated areas of Northern California. By 11:30 a.m. the smoke plume had reached Berkeley, over 225 km away. Smokey conditions persisted in California’s Central Valley and the San Francisco Bay Area until November 21. A satellite image showing the extent of smoke coverage is provided as Appendix A.

#### 2.2.2. Carr and Mendocino Complex Fires

The Carr Fire started on July 23, 2018 in the Whiskeytown district in Shasta County, California. The fire spread quickly over the next few days and burned over 28,000 acres by the evening of July 26. It ultimately consumed nearly 230,000 acres and was not fully contained until August 30. The Mendocino Complex Fire burned over 450,000 acres over the period from July 27 to November 7, 2018. The smoke from the two fires appeared to combine over large areas in the northern portion of the Sacramento Valley as shown in a satellite image provided as Appendix A. The areas burned by these two fires and the location of the impacted AQS with nearby PA-II monitor is shown in Figure 1. These two fires are henceforth described as the Carr/MC Fires.

#### 2.2.3. Pole Creek Fire

The Pole Creek Fire was the largest fire in Utah in 2018; it consumed over 98,000 acres in a steep mountain canyon approximately 90 km SSE from Salt Lake City. The smoke from the Pole Creek Fire exhibited sharp diurnal behavior consistent with changing wind patterns in a mountain environment. The AQS at Spanish Fork was the only station that recorded significant PM from the fire and had a PurpleAir monitor nearby (4 km) with publicly available data throughout the smoke event. The area burned and locations of AQS and PA-II monitors are shown in Figure 1; a satellite image of the smoke is provided as Appendix A.

### 2.3. Monitors Deployed at LBNL

The monitors used at LBNL are described in detail below with summary notes in Table 1. Reference PM_2.5_ data at LBNL were obtained using a Model 1405-DF Tapered Element Oscillating Microbalance with Filter Dynamic Measurement System (TEOM) (Thermo Scientific, Waltham, MA, USA). This device is approved as Federal Equivalent Method for 24 h measurements of PM_2.5_. As a check on the TEOM, five pairs of filter-based gravimetric samples were collected and analyzed over the study period using Gillian AirCon2 pumps (Sensidyne, St. Petersburg, FL, USA) drawing 10 liters per minute (l pm) through an Personal Environmental Monitor for PM_2.5_ (SKC, Eighty Four, PA, USA). The AirCon2 actively adjusts flow based on an internal sensor and reports an error if the flow deviates by more than 5% from the setting. The flow was also checked before each sample using a Gilian Gilibrator2 (Sensidyne). Gravimetric samples were collected on 37 mm diameter, 2 μm pore-size, TEFLO (Pall, Port Washington, NY, USA) PTFE filters that were equilibrated at a temperature of 19.5 ± 0.5 °C and relative humidity of 47.5 ± 1.5% for at least 24 h before weighing pre-and post-sampling. Filter weights were determined using an SE2-F ultra-microbalance (Sartorius, Goettingen, Germany).

The Model 1371 Mini Wide Range Aerosol Spectrometer (WRAS) (Grimm Aerosol Technik, Muldestausee, Germany) is designed specifically for indoor sampling and combines an optical particle sensor that quantifies particles in 31 size bins from 0.25 to 35 µm mean diameter and an electrical mobility spectrometer that quantifies particles in 10 size bins from 10 to 193 nm mean diameter. In addition to size-resolved number concentrations, the WRAS calculates PM_1_, PM_2.5_ and PM_10_ mass concentrations at 1-min time resolution based on the measured particle size distribution and an assumed particle density. The WRAS uses the same optical sensor as the Grimm Model EDM180, which is certified as an FEM. The EDM180 also includes a Nafion^TM^ dryer to reduce the potential for high humidity to cause significant particle growth when sampling outdoors.

Measurements were additionally made with two professional grade aerosol photometers that are used for industrial hygiene and research: a DustTrak II-8530 (DT) (TSI, Shoreview, MN, USA) that was combined with a model 801,850 heated inlet system and a Thermo pDR-1500 (PDR). These instruments have wide measurement ranges, starting at 1 µg m^−3^ and nominally extending to 150 and 400 mg m^−3^, respectively. Both have active flow control and filtered sheath air to keep the optical path clean. The PDR features temperature and humidity compensation (via software), and the heated inlet on the DT is intended to prevent artifacts from high relative humidity (RH). The two instruments are calibrated with Arizona test dust: A1-ultrafine for the DT, and A2-fine for the PDR. Appendix A shows the particle size distribution of the two dusts along with the average distribution of the smoke from the Camp Fire. Manuals for both monitors recommend coincident gravimetric sampling for calibrations to specific sources or environments and both offer flow-controlled internal filter collection. Based on the work of Wallace et al. [29], TSI recommends a calibration factor of 0.38 when using the DT to sample ambient air if no coincident gravimetric sample is obtained. The data reported in this paper uses the default calibration for the PDR and a calibration factor of 1.0 for the DT to avoid confusion. The PDR saved data every 10 s while the DT saved every 2 min. Notes on calibrations for the DT and PDR are found in the Appendix A.

Also deployed at LBNL were four low-cost IAQ monitors: the Air Quality Egg—2018 edition (AQE) (Wicked Device LLC, Ithaca, NY, USA) the AirVisual Pro (AVP) (IQAir, Goldach, Switzerland), the PurpleAir Indoor (PAI) (PurpleAir LLC, Draper Utah, USA), and the Indoor Air Quality Pro Station from eLichens (ELI) (Gremoble, France). The AVP uses a proprietary AVPM25 b sensor and calibration procedure. The PAI uses a single PMS1003 sensor (Plantower, Beijing, China) and directly reports its output. The AQE incorporates dual Plantower PMS5003 sensors and reports the average of the two sensors. The ELI uses a single Plantower PMS5003 and reports PM_2.5_ and PM_10_ after processing the data with a proprietary algorithm. The low-cost sensors appear to use a hybrid approach of optical particle sensing and photometry to estimate PM_2.5_. The Plantower sensors also report PM_1_ and particle number concentrations in 6 size bins (>0.3, >0.5, >1.0, >2.5, >5.0, >10 µm). The AQE and ELI saved data every minute, the AVP every 10 s, and the PAI every 80 s.

All monitors at LBNL were collocated within a 120 m^3^ laboratory housed within a single-story building with two exterior doors at opposite sides. Doors were closed during most of the data collection and there were no indoor sources; all PM_2.5_ in the room was thus infiltrated from outdoors. During three periods multi-hour periods totaling 26 h, the two doors were opened to increase indoor PM_2.5_ to be closer to outdoor levels. The room was not thermally conditioned. Daily high temperatures outdoors at LBNL varied from 13.2 to 19.6 °C and overnight low temperatures were 6.9 to 14.7 °C during the two weeks of the Camp Fire. Room high temperatures varied from 18.1 to 24.3 Dynamic Measurement System (TEOMC and lows varied from 14.8 to 17.8 °C. The median temperature difference (inside to outside) was 3.9 °C. The outdoor air exchange rate (AER) was not measured directly during smoke monitoring; based on prior assessments we estimate that AERs were approximately 0.5 h^−1^ or lower (refer to discussion in the Appendix A) when the door was closed.

### 2.4. Data from Regulatory Air Quality Monitoring Stations

Data from air quality monitoring stations (AQS) in California were obtained from the Air Quality and Meteorological Information System (AQMIS) maintained by the California Air Resources Board [40]. The regulatory network in California uses beta attenuation monitors (BAM) models 1020 and 1022 (Met One Instruments; Grants Pass, OR, USA) to record hourly PM_2.5_. A BAM draws air through a size-selective inlet to set the PM mass fraction being measured (e.g., PM_2.5_ or PM_10_) then through a filter tape to collect sample. Collected particles change the attenuation of beta rays passing through the filter tape proportionally to the mass of particles collected. The change in mass over the measurement time interval is divided by the sample air volume to calculate PM_2.5_ concentration. 

Data from the Spanish Fork monitoring site that was impacted by the Pole Fire in Utah was obtained from the Utah Department of Environmental Quality website [41]. The FEM monitor operating at this site was a Thermo Scientific Model 5030i Synchronized Hybrid Ambient Real-time (SHARP) particulate monitor. The SHARP monitor combines an optical particle counter with a beta attenuation instrument. The optical portion of the instrument provides a data stream with high temporal resolution, and the beta attenuation provides a mass measurement to dynamically adjust the optical instrument and provide accurate time resolved PM mass concentration.

### 2.5. PurpleAir Network

The PurpleAir PA-II monitor features two Plantower PMS5003 sensors, electronics, and software to enable quick connection to the web via wifi-all packaged in a 4″ PVC cap with an outdoor power supply for weather protection. When setting up a new device the user is prompted to set the geographic location and whether it is indoors or outdoors, with outdoors as the default. PurpleAir provides a real-time, map-based data display (https://www.purpleair.com/map#1.1/0/–30) and enables downloads of data from its server. Device owners have the option of making the data publicly available or accessible only to users whom they designate. To our knowledge, PurpleAir had by far the largest distributed network of PM sensors with publicly viewable data deployed around California at the time of the Camp Fire. We also observed expansion of the network in terms of number of monitors and spatial coverage during the fire. 

The default setting on the PurpleAir map presents data as the US EPA PM_2.5_ AQI calculated from the PM_2.5_ concentration reported by each Plantower sensor. The online map allows the user to display any of the other data streams provided by the sensors and other AQI-type values. At the time of the wildfires (and still on 07-May-2020), the site offered two “conversions” to adjust PM_2.5_ concentrations and corresponding AQI values. The site attributes an “AQandU” calibration (0.778 * PA + 2.65) to a long-term University of Utah study in Salt Lake City and an “LRAPA” calibration (0.5 * PA − 0.68) to a Lane Regional Air Pollution Agency study of PA sensors. The University of Utah evaluated Plantower sensors measuring ambient air in Salt Lake City during all seasons of the year [42,43]. The LRAPA adjustment is from a winter study performed in a region of Oregon that has widespread use of wood combustion heating and ambient PM_2.5_ that is predominantly composed of wood smoke when at its highest levels [44].

### 2.6. Identification of Paired PA-II and Regulatory AQ Monitoring Data

To conduct the analysis described herein, we manually searched the PA map to identify PA-II monitors within ~5 km of an AQS site that reported PM_2.5_ on an hourly basis during any of the fires examined herein. To assess if PA-II and AQS data were appropriately paired, we considered local topography such as the presence of valleys or mountains that could result in the PA and AQS seeing different air masses and also viewed data to confirm basic synchronicity of trends. For the Camp Fire we identified 53 PA-II monitors in the vicinity of 12 NorCal AQS sites. From these 53 PA-II devices, we downloaded data from 97 sensors that reported data that appeared valid based on the review described in the next section. The median distance between the AQS and PA-II monitors was 2.7 km, the interquartile range was 1.1–4.6 km, and the full range was 0–11.6 km. For both the Carr/MC and Pole Creek Fires we found a single PA-II monitor and nearby AQS combination. For the Carr/MC Fire the AQS was approximately 50 km from the fire, and the PurpleAir was co-located with the site. The AQS selected for the Pole Creek Fire was approximately 35 km from the fire and the PA-II monitor was ~4 km from the AQS.

### 2.7. Analysis of Data from PA-II Monitors

The PA-II monitors report data at 80 s resolution. Many of the devices had occasional data gaps, presumably due to wifi connectivity issues. AQS data are provided at 1 h resolution. The unadjusted cf_1 data stream reported by the Plantower sensors to the PA-II monitor were used to calculate 1 h averages and both PA-II and AQS data were aggregated to 4 h averages to account for the sites not being precisely co-located. Correlations and adjustment factors were evaluated with the 4-h data streams. (The cf_1 and cf_atm data streams were switched in PA-II reporting at the time of the fire [45], but we have subsequently confirmed that the stream used in this study was the stream that is currently labeled as cf_1). 

Most PA-II devices provide data for each of the two onboard sensors and occasionally the sensors read significantly different results. As a quality assurance screen, we reviewed data from all of the individual sensors of PA-II units. This was done by plotting the time series of the raw values along with the AQS data on the same plot, visually identifying the outliers. We flagged any sensor that substantially diverged from other nearby sensors, including from the same monitor. We also identified devices that appeared to be indoors but were marked as outdoors, indicated by the two sensors of the device agreeing closely and being well below the group or having occasional peaks (presumably from indoor PM emissions) that did not appear in the other outdoor devices. Faulty sensors and presumed indoor units were removed from the analysis. 

## 3. Results

### 3.1. Reference Measurements at LBNL

Over the course of the fire, PM_2.5_ as measured by the TEOM inside the lab at LBNL averaged 47.3 µg m^−3^ or roughly half of the event-averaged concentrations of 93.2 and 93.9 µg m^−3^ at air quality stations in Berkeley (4.5 km to the West) and Oakland-West (7.3 km to the Southwest). Appendix A presents the time concentration profiles of hourly-averaged PM_2.5_ for these sites. Duplicate filters provided consistent results and generally agreed with the TEOM (Appendix A). Some difference is expected since the TEOM sampling sequence did not perfectly align with some filter sample intervals. 

### 3.2. Measurements with Low-Cost, Professional and Research Monitors at LBNL

Time series of PM_2.5_ concentrations reported by all monitors deployed at LBNL are shown in Figure 2, which shows that all tracked with the TEOM and all but the WRAS substantially over-reported PM_2.5_ throughout the event. Responses relative to TEOM varied for both professional-grade and low-cost monitors. For each monitor, we calculated the statistics of relative response (device reported PM_2.5_ divided by TEOM PM_2.5_) using event-integrated and 4-h average data, with results provided in Appendix A. The event-integrated mean and median 4-h response factors were closest to unity for the WRAS and farthest for the DustTrak. 

The DustTrak and pDR-1500 both use Arizona test dust as their calibration aerosol; and the devices nevertheless provided different responses. The DustTrak calibration is based on A1 Ultrafine dust, with mass median diameter (mmd) in the range of 3 to 5 µm. The PDR is calibrated with Arizona test dust A2 Fine, with mmd of 8 to 10 µm. These calibration aerosols have very different size distribution and optical properties than wildfire smoke, as shown in Appendix A. The two devices also use light sources of different wavelengths and measure at different scattering angles.

### 3.3. Adjustment Factors Based on LBNL Measurements

Low-cost and professional monitor measurements were related to actual PM_2.5,_ as measured by the TEOM, by determining the best fit parameters for linear equations with zero or non-zero intercepts. While prior studies have reported substantial non-zero intercepts when using low-cost monitors to measure ambient aerosols [46,47,48,49], we found that for the wildfire smoke, the slopes were very similar with zero or non-zero intercepts (see Appendix A). Based on this, we subsequently report adjustment factors as simple scalars with no offset. 

Adjustment factors (AF) to translate the PM_2.5_ reported by each instrument to the PM_2.5_ reported by the TEOM (i.e., TEOM PM_2.5_/device PM_2.5_) were calculated for each 4-h interval of data and for the entire event. Summary results, provided in Figure 3, show that AFs varied across devices and also over time for each device. The DT had the least variability in part because it was used for only a few days. The WRAS reported concentrations closest to the TEOM with a median AF of 0.85. Median AFs for the monitors with low-cost sensors varied from 0.42 to 0.60.

An example of applying the AFs to adjust time-resolved data is provided in Figure 4. The bottom panel shows the AFs for 4-h average data over the course of the fire for the three AVP units. The middle panel shows that the adjusted time-series (using the median of the 4-h AFs across the event) closely match the TEOM and the top panel shows that residual errors were almost all between −30% and +20%. The same plot is provided for the PAI as Appendix A. For both the AVP and PAI, three units of each device agreed closely throughout the fire. 

### 3.4. Measurements of PM_2.5_ and Adjustment Factors in Northern California and Utah

Table 2 provides summary data from the 12 NorCal air quality monitoring stations that had nearby PA-II monitors at the time of the Camp Fire. The AQS sites varied in distance from the town of Paradise, which was the focal point of the fire. The positions of the AQS sites relative to the burned area are shown in Figure 1. Appendix A provides the number of PA-II monitors and reporting sensors. Given the large distances, there was relatively small variation of PM_2.5_ across sites. Sacramento was closest and had the highest mean concentration. Davis was nearly the same distance from the fire but had among the lowest concentrations, possibly due to it being on the edge of the plume for much of the event (based on satellite images). Sites between Vallejo and San Francisco showed remarkable consistency with a relative standard deviation in the mean event concentration of 3.4%. Mean concentrations at the two furthest sites, San Jose and Redwood City, were about 15% lower than most of the other sites. 

Table 2 also presents linear fits, with zero and non-zero intercepts, relating 4-h intervals of PA-II sensor and nearby AQS data. Table 2 presents AFs calculated for all AQS sites as the medians of all available 4-h AFs from sensors near the sites. Distributions of daily AFs across sites over the course of the fire are presented in Figure 5.

The median AF for the 12 AQS sites varied between 0.42 and 0.49 for the first 9 days, rose to 0.57–0.58 on Days 9–10, and declined over the last few days. To assess variance, we consider the relative median absolute deviation (RMAD) statistic, which is analogous to the relative standard deviation (RSD). The median RMAD for all the 4-h average AFs—including all days and all sensors —nearby to individual AQS sites was 12% with an IQR of 8–18% (Appendix A). An event specific adjustment factor (ESAF) for the Camp Fire was calculated as the median of all daily AFs for all AQS sites, providing a value of 0.485. Examples of time-series adjusted with the regional ESAF are provided for three sites in Figure 6. Whereas a prior study by Stampfer et al. [50] reported non-linear response for Plantower sensors when PM_2.5_ was above 25 µg m^−3^, the PA data used in this study linearly tracked with PM_2.5_ to above 200 µg m^−3^ as shown in Appendix A. Since deviation from the linear relationship occurred only at very high levels, which were infrequent (Appendix A) the simple linear adjustment was applied.

Applying the regional adjustment factor to PA-II measurements throughout the area substantially reduced errors relative to default PA-II output. Figure 7 presents the summary distributions of residual errors of 4-h average data across all 12 AQS sites over all days of the smoke event using unadjusted values along with three different adjustment factors. The median unadjusted error was +102% with an IQ range of 74–133%. Using the regional ESAF from the Camp Fire produced an interquartile range of roughly ±15% in the residual error. The LRAPA adjustment available on the PurpleAir map at the time of the fire produced very similar results as using the regional AF calculated in this study. Using the AQandU correction resulted in concentration estimates that were 61% off at the median with an IQ range of 38–85%. 

We applied the same analysis to the single PA-II and AQS combination that was available during the Carr/MC Fire. This site was heavily impacted by smoke from July 26 through August 12, 2018. Figure 8 shows the errors for the PA-II compared to the Red Bluff AQS using both unadjusted and adjusted data. The unadjusted PA-II monitor had a median error of +135% with an IQ range of 112–153%. When data were adjusted using the regional Camp Fire AF, median residual errors were 13% (3–22% IQR). When the Carr/MC data were adjusted with the AF determined for that fire, the IQR of the residual error was −9.5 to 7% (with median of 0%).

The Pole Creek Fire intermittently pushed smoke out into the Utah Valley from September 13 through September 24, 2018. Appendix A shows the AQS and PA-II data—both unadjusted and adjusted using the Camp Fire AF, along with errors in the adjusted data. The AQS data showed a strong diurnal pattern with clean air in the evening around sunset and smoke starting to enter the valley by midnight and peaking a few hours after sunrise. When performing the analysis of PA-II and AQS data for this site, we focused only on the periods impacted by the smoke event by excluding any 4-h intervals where the AQS was <35 µg m^−3^. When smoke was present, errors in the unadjusted PA-II PM_2.5_ were +111% (66–212% IQR). Errors were reduced to 1.8% (−20 to +51% IQR) when using the Camp Fire regional ESAF. When adjusting data with the AF determined for the Pole Creek Fire, the IQR of the residual error was −21 to +48%.

Figure 9 shows the adjustment factors for the three fires. Three distributions are shown for the Camp Fire: regional ESAFs from the 12 AQS sites, the AFs determined at the Berkeley AQS, and the AFs measured inside the lab at LBNL. While AFs for the Berkeley AQS had a similar median and range as those from sites across the region, AFs for the infiltrated PM_2.5_ at LBNL were lower. This could be due to several factors. First is the possibility that the particle size distribution of PM_2.5_ inside the lab was different than outside around the region. Since PM_2.5_ levels inside were lower by about a factor of two, there was clearly some loss of particles relative to outside. It is well established that the size distribution can change as penetration and deposition rates vary with particle size, leading to uneven losses across the range of particle sizes [51,52,53]. Another possible factor is that the reference instrument at LBNL was a TEOM while the air monitoring stations in Berkeley and elsewhere in California used BAM instruments. A third potential factor is the different versions of the Plantower sensor used in the PAI devices inside at LBNL (which use PMS1003 sensors) and the PA-II, which uses the PMS5003 sensor. The two sensors have the same nominal specifications and appear to use the same electronic components but they have different internal flow pathways. The Carr/MC Fire required a larger adjustment than was needed for the Camp Fire. The variability in the Pole Creek AFs may be impacted by the diurnal variability (which was still present even when analyzing only those intervals with PM_2.5_ > 35 µg/m^3^), coupled with the fact the PA-II was ~4 km away from the AQS for this site. As the plume was moving either in or out it was hitting the PA-II and AQS at different times. By contrast, for the California fires the plumes were present for days at a time reducing spatial and temporal variability.

### 3.5. Impact of Adjustments on Air Quality Index Estimates

Adjustments to the reported concentrations translate to major changes to the associated AQIs reported for PA-II monitors. Figure 10 provides examples for PA-II monitors nearby to three monitoring sites during the Camp Fire in Northern California. For each site, three time series of 4-h AQI values are presented. The top bar is calculated from unadjusted PA-II readings. The middle is calculated from PA-II readings adjusted with the regional ESAF. And the bottom row is the 4-h AQI calculated from AQS data. At the Sacramento site, which was the closest of the three to the fire at 135 km away, unadjusted PA-II data indicated an AQI of “very unhealthy” or “hazardous” for 83% of the smoke event. Adjusted data indicated an AQI in these categories 31% of the time, which is similar to the 30% of time that the regulatory monitor reported AQI in these categories. Unadjusted sensor readings indicated the correct AQI category 14% of the time whereas the adjusted PM_2.5_ provided the correct AQI category 84% of the time. At the San Pablo site (210 km from the fire), unadjusted PA-II indicated “very unhealthy” or “hazardous” for 59% of the event duration, which was much higher than the 17% of time that PM_2.5_ measurements from the AQS indicated those AQI categories; adjusted PA-II data indicated these AQI categories of concern 10% of the time. Overall, the AQI category calculated from unadjusted PA-II data matched the AQS AQI only 29% of the time while the adjustment resulted in the correct AQI range 65% of the time. At the San Jose site (270 km from the fire), unadjusted PA-II indicated “very unhealthy” or “hazardous” for 34% of the event duration, whereas the 0% of time for the ESAF-adjusted data was similar to the 2% for the AQMS. Unadjusted data predicted AQI 47% of the time and adjustment led to the correct AQI category 66% of the time. 

Despite the overall improvements in AQI predictions, it is important to note that adjusted values sometimes predicted a lower AQI hazard level than indicated by the nearby AQS.

### 3.6. Impact of Environmental Conditions on Adjustment Factors

PA-II adjustment factors were similar for the three fires, which occurred under varied seasonal conditions: the MC/Carr fire occurred in early summer (June), the Pole Fire occurred in late summer (September) and the Camp Fire occurred in late fall (late November). To assess the relationship between AFs for wildfire smoke and those occurring at the same sites through the year, we analyzed data from all of 2019 at measurement sites impacted by the Camp and MC/Carr fires in 2018. We first screened the data to look only at 4-h intervals with PM_2.5_ above 12 µg/m^3^. Appendix A show distributions of temperature, RH, and PM_2.5_ for these intervals of elevated PM_2.5_ during each of the three main seasonal conditions (winter = December through February; summer = May through September; and shoulder = all other months), and for the period of each wildfire in 2018. During the Camp Fire, RH spanned a range similar to the shoulder seasons and temperatures were closer to those during winter. Consistent with this, the AF distribution was midway between the fall and winter distributions during periods of elevated PM_2.5_. In Red Bluff, temperature and RH during the wildfire were similar to 2019 summer periods with elevated PM_2.5_, but the distribution of AFs closely matched those during winter elevated PM_2.5_. One possible reason for the similarity of wintertime and wildfire smoke AFs is that a substantial fraction of elevated winter PM_2.5_ at both T-street and Red Bluff AQS sites may be associated with smoke from home heating wood combustion. The effect of environmental conditions is explored further in Appendix A, which show AFs as a function of RH, by season, at the two sites. In both cases the AFs for wildfire smoke do not change much over a very broad range of RH conditions. AFs also do not vary with RH during the winter. By contrast, AFs vary with humidity for the summer and shoulder seasons. Collectively these results provide support for the hypothesis that the AFs identified for smoke events are directly related to the characteristics of the smoke and do not vary greatly with environmental conditions. 

## 4. Discussion

Prior research indicates varying responses of optical particle sensors to smoke generated from biomass combustion. In the most extensive and directly relevant study, McNamara et al. reported ratios of DustTrak 8520 and 8530 models to gravimetric, FRM and FEM measurements of varied instances of PM_2.5_ from wood smoke [54]. Inside homes with wood stoves, which had PM_2.5_ elevated from loading and stoking events, ratios of DT to gravimetric measurements were 1.60 ± 1.05 (mean AF = 0.63) across 43 sampling periods with mean (SD) gravimetric PM_2.5_ = 30.7 (34.7) µg/m^3^. DT to gravimetric ratios were 1.59–1.70 (AF = 0.63–0.59) for sampling in a University of Montana laboratory during three 24-h periods impacted by forest fire smoke (gravimetric PM_2.5_ = 11.3, 21.2, 55.3 µg/m^3^). For wintertime ambient sampling of PM_2.5_ impacted by wood smoke in Libby, Montana, DT to BAM ratios were 1.43 ± 0.61 for BAM-reported PM_2.5_ of 24.6 ± 8.0 µg/m^3^. Dacunto et al. reported fireplace wood smoke correction factors (equivalent to AF) of 0.44–0.47 for the TSI Sidepak photometer, which is similar to the DT [55]. The previously reported AFs for wood and wildfire smoke are higher than the DT AF of 0.25 measured at LBNL for infiltrated wildfire smoke. These differences are presumed to result from variations in composition and size distributions of the measured aerosols, which had varied generation, ageing, and environmental conditions. Using an early generation of the pDR instrument, a U.S. Department of Agriculture study reported an AF of 0.53 for smoke generated in a fire laboratory [56]; the same AF was measured in our current study for the modern version of the pDR for infiltrated wildfire smoke.

There are limited published data on wood or wildfire smoke AFs for low-cost monitors. The Lane Regional Air Protection Agency (LRAPA), which is in a region of Oregon that is impacted by wood smoke from home heating, compared PA-II monitors to their network of FEMs and reported an AF equation of 0.5 * PA(PM_2.5_)–0.66 [44]. This is offered as a checkbox conversion on the PA map and closely aligns with the Camp Fire regional AF of 0.48. A long-term study of Plantower 1003 and 5003 sensors by the University of Utah reported equations relating hourly averaged individual sensor readings to a collocated TEOM 1405-F FEM-approved monitor by season [42]. During the months of Jun–Oct, which the authors described as “wildfire season” because of several fire events that occurred during that period, slopes of the linear fits were in the range of 1.33–1.48 (roughly corresponding to AFs of 0.68–0.75). However, since there were relatively few hours with PM_2.5_ > 40 µg/m^3^ over the season and the analysis did not break out the fits during the few significant wildfire events, the reported fits are not directly appropriate to adjusting Plantower sensor data during wildfire events with high PM_2.5_ (e.g., >40 µg/m^3^).

## 5. Conclusions

Low-cost air quality monitors can be used to accurately estimate hyper-local concentrations, regional dispersion, and health risk of PM_2.5_ from wildfire smoke if appropriate device-specific adjustment factors are applied. Data from the existing network of outdoor PurpleAir II monitors is currently available with substantial coverage in many locations throughout the world. While the default PA-II response substantially over-reports wildfire smoke PM_2.5_, the data can be scaled using the adjustment factor of 0.48 determined for the Camp Fire in Northern California, leading to substantially more accurate air quality index estimates. Based on measurements at LBNL of infiltrating smoke PM_2.5_, it appears that both professional grade photometers and other monitors using Plantower low-cost optical PM sensors also substantially over-report wildfire PM_2.5_ values. A simple multiplicative adjustment factor can bring the low-cost monitor response much closer to the PM_2.5_ and AQI that would be reported by a regulatory monitor at the same location. Wildfire smoke AFs can vary across locations and over time during a fire event and the median AFs from one event may differ somewhat from those at other events. Yet even with these variations, application of a global AF can reduce bias from roughly a factor of two to 20–30% or less. It is also possible to apply short term adjustment factors by comparing the previous several hours of data from a deployed sensor to that form a nearby AQS if one is available. 

## Figures and Tables

**Figure 1 sensors-20-03683-f001:**
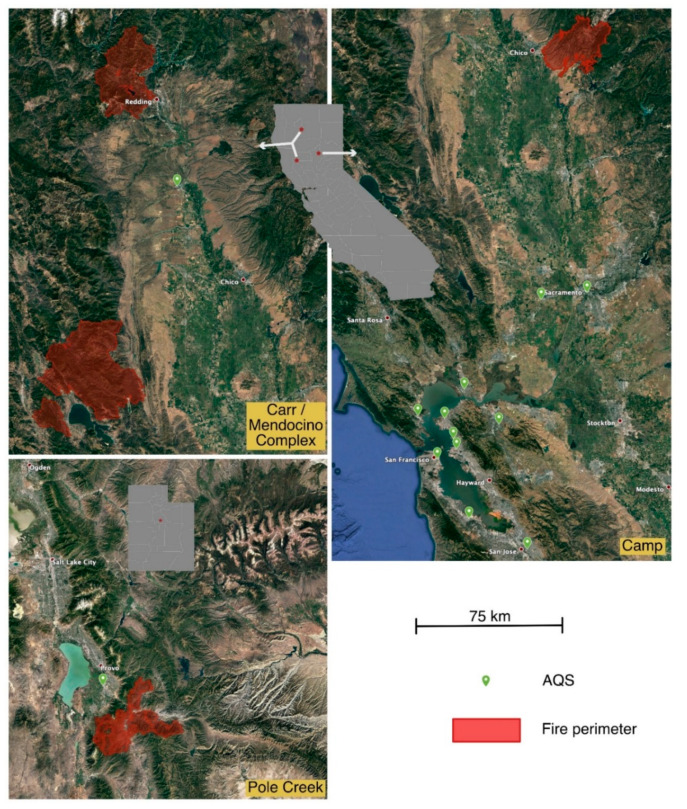
Areas burned (red) and locations of air quality monitoring stations (green dots) reporting hourly PM_2.5_ that also had nearby PA-II monitors. Gray images are the states of California (top) and Utah (bottom) and red dots show the locations of the detailed maps with the states. The 75 km scale marker applies to all three detailed images. Satellite images were obtained via Google Earth. Fire extent data from https://fsapps.nwcg.gov/googleearth.php.

**Figure 2 sensors-20-03683-f002:**
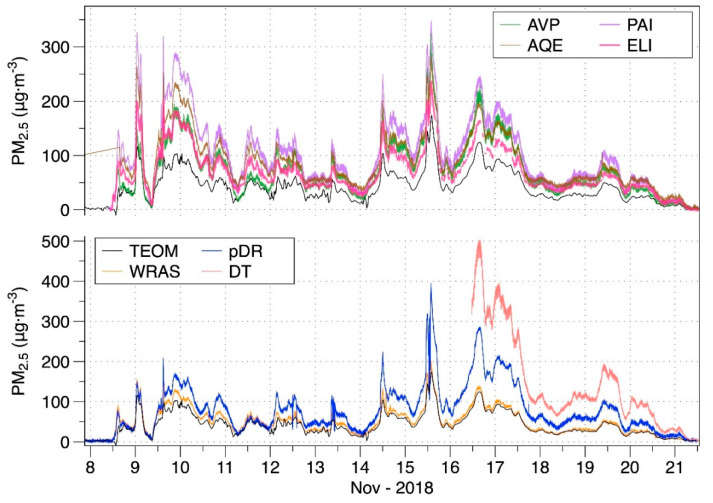
Time series of PM_2.5_ as reported by all monitors tested at LBNL. The TEOM is a U.S. Federal Equivalent Method, and thus considered as the reference data. The top group are the monitors utilizing low-cost sensors and the bottom group are professional research grade monitors, with the TEOM reference measurement shown with both groups. Refer to Table 1 for monitor descriptions.

**Figure 3 sensors-20-03683-f003:**
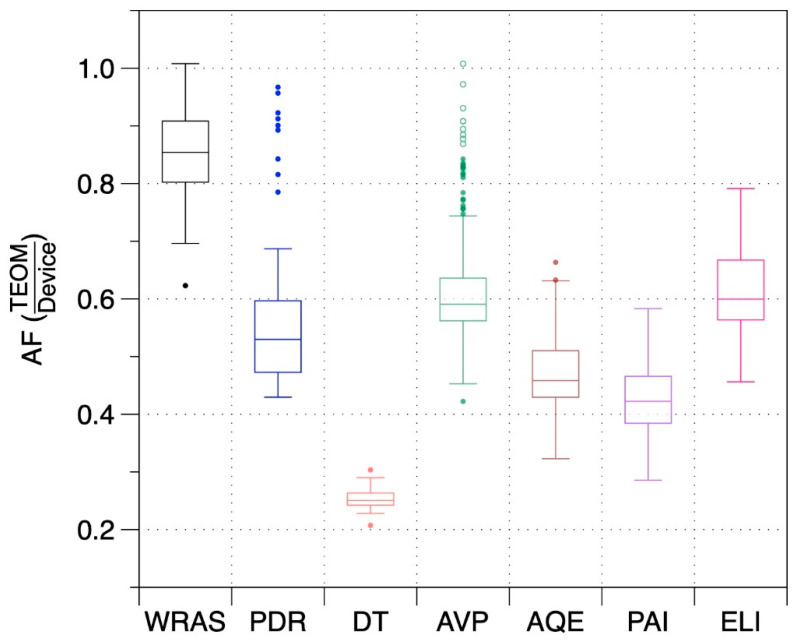
Distributions of 4-h average adjustment factors determined from measurements in a naturally ventilated lab over 13 days of elevated smoke from the Camp Fire.

**Figure 4 sensors-20-03683-f004:**
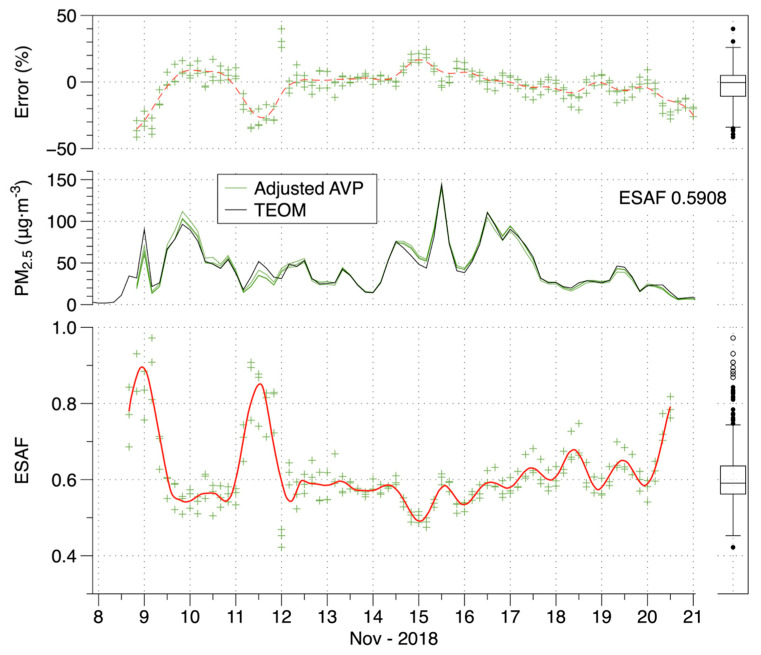
Event-specific adjustment factors (ESAFs), adjusted data and error of adjusted data (relative to co-located TEOM PM_2.5_) for 4-h average AVP measurements in a naturally ventilated lab over 13 days of elevated smoke from the Camp Fire.

**Figure 5 sensors-20-03683-f005:**
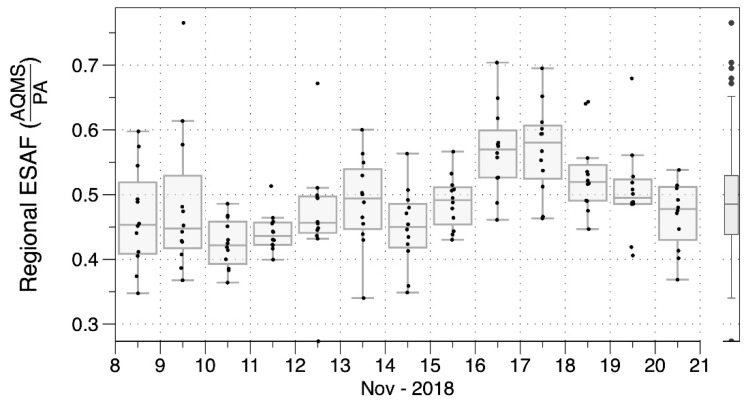
Variation in regional event-specific adjustment factor over time, shown as distributions of daily ESAFs required for PA-II monitors to align with measurements at 12 regulatory monitoring sites in Northern California during the Camp Fire in November 2018.

**Figure 6 sensors-20-03683-f006:**
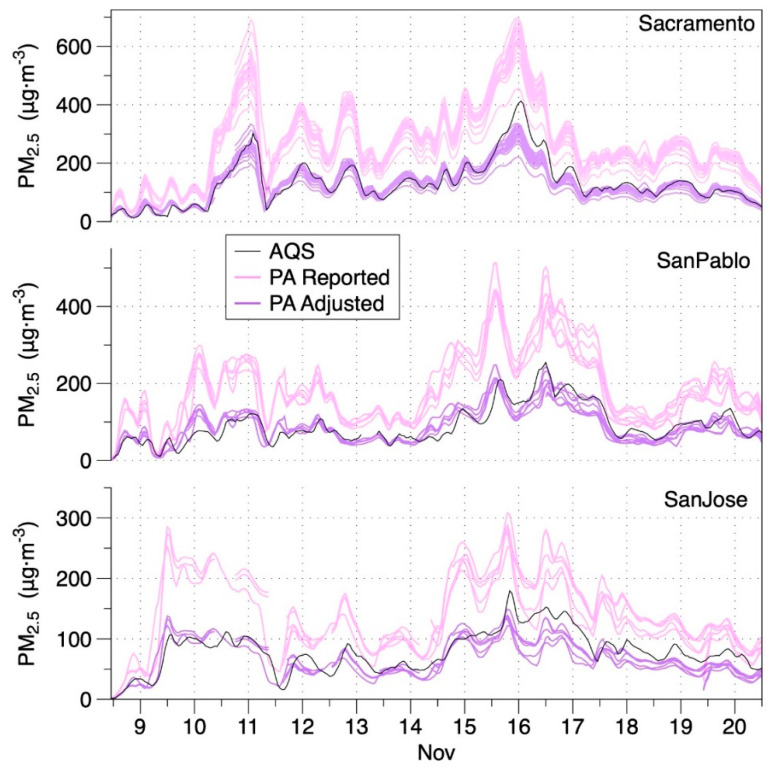
PM_2.5_ concentrations reported by regulatory air quality monitoring stations (AQS) and all valid sensors of PurpleAir PA-II monitors nearby to three AQS sites during the Camp Fire in Northern California in November 2018. Pink data are unadjusted and purple data are adjusted using the regional event specific adjustment factor of 0.485.

**Figure 7 sensors-20-03683-f007:**
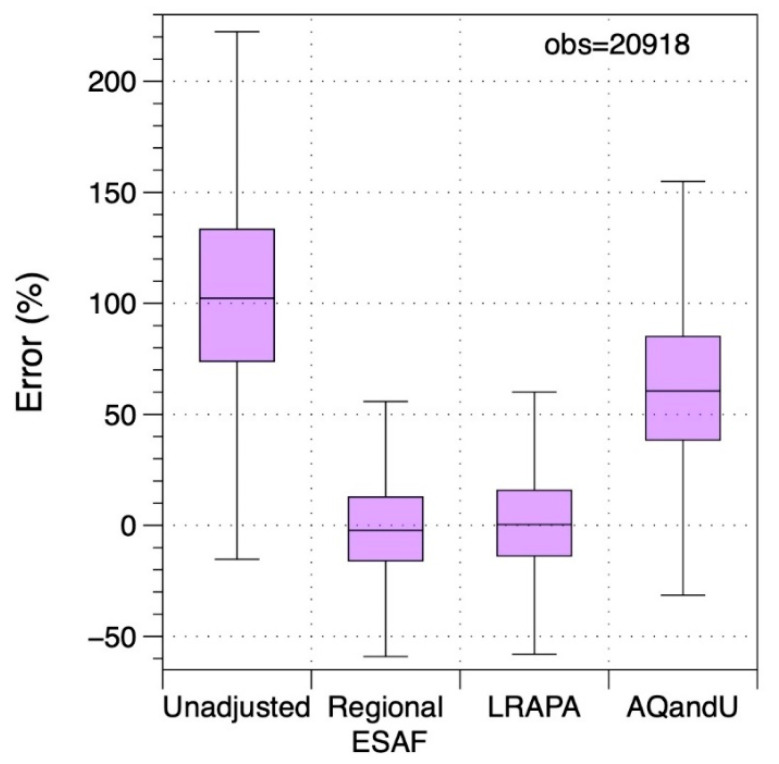
Distributions of errors for 4-h average PM_2.5_ concentrations for Northern California PA-II monitors compared to regulatory monitors for the Camp Fire using different adjustment factors.

**Figure 8 sensors-20-03683-f008:**
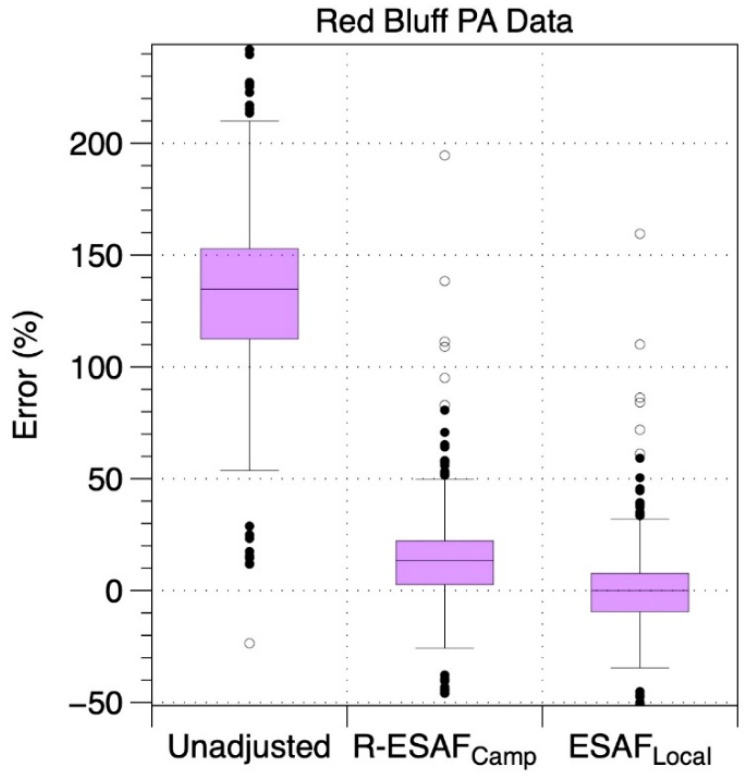
Distributions of errors for 4-h average PM_2.5_ concentrations for Red Bluff PA-II direct readings and after applying the regional event specific adjustment factors from Camp Fire (R-ESAF) or those derived for the local Carr/Mendocino Complex Fire.

**Figure 9 sensors-20-03683-f009:**
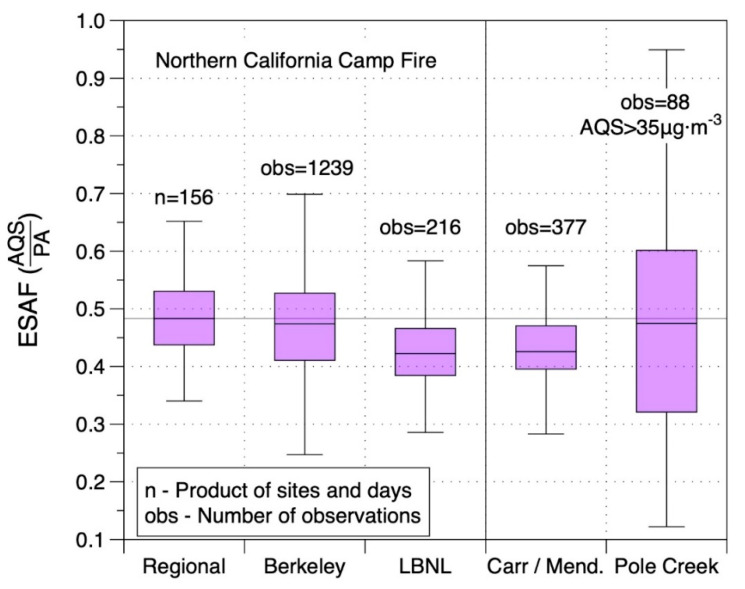
Event specific adjustment factors (ESAFs) calculated for PA-II monitors using data from three wildfire smoke events in 2018. The LBNL results are for indoor monitoring of infiltrated PM_2.5_ in a lab with an estimated outdoor air exchange rate of 0.5 h^−1^.

**Figure 10 sensors-20-03683-f010:**
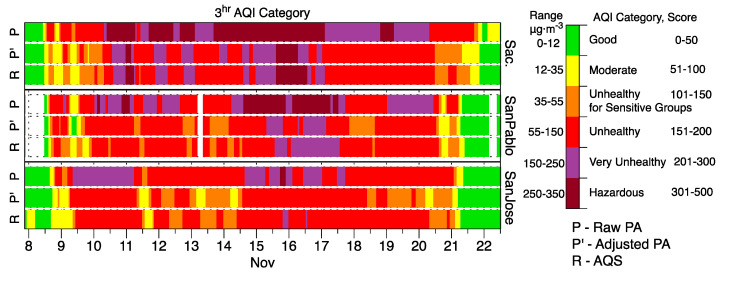
Comparison of Air Quality Index (AQI) values calculated using unadjusted data from PA-II monitors and the same data after correction with the regional, event specific AF to AQIs from PM_2.5_ data at regulatory monitoring sites in Northern California during Camp Fire.

**Table 1 sensors-20-03683-t001:** Descriptions of monitors used in this study.

ID	Device	Data	Particle Sensor(s) and Specifications for PM_2.5_	Calibration and Quality Assurance Information Provided by Manufacturer
AQE	Air Quality Egg 2018 version	1 min	Two Plantower PMS5003 ^1^ Effective range: 0–500 μg/m^3^ Max. range: ≥1000 μg/m^3^ Max. consistency error: 0~100 μg/m^3^: ±10 μg/m^3^ 100~500 μg/m^3^: ±10%	https://airqualityegg.com/home Reports mean PM_2.5_ and PM_10_ of the two sensors. Each unit checked for consistency with other devices before shipping by exposure to incense smoke in a small room.
AVP	IQAir AirVisual Pro	10 s	AirVisual AVPM25b Effective range: 0–1798 μg/m^3^	https://www.airvisual.com/ Sensors calibrated through automatic process in controlled chamber, using distinct aerosols for PM_1_, PM_2.5_, PM_10_ using Grimm 11-A.
PAI	PurpleAir Indoor	80 s	Plantower PMS1003 Same specification as PMS5003	https://www.purpleair.com/sensors Data direct from sensor: PM_1_, PM_2.5_ and PM_10_ in μg/m^3^, number density (#/0.1 L) of particles larger than the following optical diameters: 0.3, 0.5, 1.0, 2.5, 5.0, 10 μm.
ELI	eLichens Indoor Air Quality Pro Station	1 min	Plantower PMS7003 Same specification as PMS5003	https://www.elichens.com/elsi-indoor-air-quality-station Each station individually calibrated against regulatory AQ stations meeting EU standards. Data are adjusted in real-time for environmental conditions.
PA-II	PurpleAir II (outdoor)	80 s	Two Plantower PMS5003	https://www.purpleair.com/sensors Same as PAI.
TEOM	Model 1045-DF Tapered Element Oscillating Microbalance with Filter Dynamic Measurement System	12 min	Range: 0 to 1,000,000 μg/m^3^ Resolution: 0.1 μg/m^3^, Precision: ±2.0 μg/m^3^, 1-h avg	https://www.thermofisher.com/ Approved Federal Equivalent Method U.S. EPA PM-2.5 Equivalent Monitor EQPM-0609-182.
WRAS	Model 1.371 Mini Wide-Range Aerosol Spectrometer	1 min	Combined electrical mobility instrument with optical particle spectrometer. Range: 0.1 μg/m^3^–100 mg/m^3^ Electrical mobility sensing: 10 bins in range 10–193 nm, Optical sensing 31 bins in range 0.253–35 μm	https://www.grimm-aerosol.com Optical spectrometer calibrated using class I reference with NIST-certified, mono-disperse polystyrene latex (PSL) particles. Electrical sensor calibrated using GRIMM model 7811 with poly-disperse aerosol of particles with diameters of ~5 nm to ~300 nm generated from NaCl solution. Aerosol is dried and diffusion-neutralized. A Differential Mobility Analyzer (DMA) provides narrow size distributions simultaneously to the sensor and a reference Faraday cup electrometer.
PDR	Thermo pDR-1500	10 s	Laser optical photometer Range: 0.001–400 mg/m^3^ Precision: larger of ±0.2% of reading or ±0.0005 mg/m^3^. Accuracy: ±0.5% reading ±precision	https://www.thermofisher.com/ Traceable to SAE Fine Test Dust.
DT	TSI DustTrak II-8533	2 min	Laser optical photometer Range: 0.001 to 150 mg/m^3^ Flow Accuracy: ±5% factory setpoint Internal flow controlled	https://tsi.com/home/ Calibrated with ISO 12103–1, A1 Ultrafine Test Dust.

^1^ Plantower documentation describes the analytical method as follows: “…collect scattering light in certain angle [and] obtain the curve of scattering light change with time. [By microprocessor, calculate] equivalent particle diameter and the number of particles with different diameter per unit volume based on MIE theory. Product documentation also reports “endurance max error” after 720 h of operation: as ±15 μg/m^3^ for 0~100 μg/m^3^ and ±15% for 100~500 μg/m^3^.

**Table 2 sensors-20-03683-t002:** Calculated linear fitting parameters with/out a zero offset and adjustment factors to quantify wildfire smoke PM_2.5_ based on comparisons of 4-h means from PurpleAir PA-II monitors nearby to Northern California regulatory air quality monitoring stations during the Camp Fire in 2018.

		PM_2.5_ (µg m^−3^), 4-h avgs.			Linear Fits of 4-h avg Data Relating PA-II to AQS			Adjustment Factors Based on 4-h Ratios of AQS/PA-II				
AQS Site	Distance (km)	Mean	10th	90th	Slope, Zero Intercept	Slope	Intercept	Mean	SD	Median	10th	90th
Sacramento	133	134	47	239	0.498	0.510	−3.2	0.509	0.106	0.487	0.393	0.626
Davis	137	82	15	169	0.425	0.410	3.0	0.411	0.165	0.419	0.293	0.558
Vallejo	192	92	38	175	0.490	0.490	0.0	0.490	0.142	0.490	0.373	0.641
Concord	206	87	33	160	0.474	0.445	4.9	0.494	0.167	0.494	0.341	0.758
San Pablo	210	93	47	162	0.504	0.455	9.7	0.499	0.108	0.488	0.397	0.630
San Rafael	213	89	46	153	0.495	0.505	−2.0	0.631	0.255	0.635	0.439	1.092
Berkeley	219	93	54	164	0.459	0.423	6.9	0.464	0.089	0.472	0.375	0.572
Oakland-West	224	94	51	161	0.532	0.509	3.9	0.465	0.079	0.458	0.383	0.563
Oakland-Laney	226	91	53	157	0.459	0.437	4.5	0.530	0.088	0.528	0.423	0.627
San Francisco	232	93	45	156	0.511	0.498	2.5	0.504	0.160	0.520	0.318	0.692
Redwood City	258	74	33	122	0.446	0.387	8.3	0.451	0.162	0.449	0.314	0.607
San Jose	270	80	40	126	0.574	0.536	4.9	0.482	0.140	0.484	0.353	0.593

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
