# Peer review of "Wildfire Smoke Adjustment Factors for Low-Cost and Professional PM2.5 Monitors with Optical Sensors"

_sensors, 2020, doi:10.3390/s20133683_

Round 1

Reviewer 1 Report

This work is really well done. It is rare to see tidy published works on AQ sensors. The results are well presented and their synthesis is excellent. 

I found the outcomes of AQ sensors in line with the reference systems. The good results obtained by the use of the Purple Air sensor also confirm the good quality of this sensor when compared to other AQ sensors for PM2.5 commercially available. I would like to ask the authors to spend few words about the fact that the purple air sensor showed good performance and this is in line with what reported in the work https://doi.org/10.3390/atmos10090506 (Atmosphere 2019) about comparison of AQ sensors.

Therefore I suggest publication of this work with no revisions.

Author Response

Thank you for the positive feedback!

Reviewer 2 Report

This paper is well written and will be of interest to many people trying to use low-cost and professional PM2.5 monitors during wildfire smoke events. I have primarily minor comments that I hope may strengthen the text and improve reproducibility.

Comments:

  1. Plantower sensors report PM2.5 with two corrections cf_1 and cf_atm. There is much confusion surrounding these outputs for PurpleAir since the labels were switched until late 2019. This is described in Tryner et al. Can you please specify which cf you are using and whether you are using the current naming convention or the pre 2020 naming convention? This is likely something to define for the Plantower based sensors other than the PurpleAir as well.
    1. Tryner, Jessica, Christian L'Orange, John Mehaffy, Daniel Miller-Lionberg, Josephine C. Hofstetter, Ander Wilson, and John Volckens. 2020. 'Laboratory evaluation of low-cost PurpleAir PM monitors and in-field correction using co-located portable filter samplers', Atmospheric Environment, 220: 117067.
  2. It would likely be helpful if you added a table summarizing instruments, acronyms, type (FEM/filter/professional/low-cost etc) and timelines. This may help readers quickly reference monitor acronyms as they come up.
  3. It seems like a lot of previous work with PurpleAir and other sensors has adjusted the data using linear regression including an intercept significantly different than zero (e.g. Mehadi, Maling, Feenstra). If you have an intercept significantly different than zero it would impact your AF differently at low and high concentration. Did you consider this when doing this analysis?
    1. Malings, Carl, Rebecca Tanzer, Aliaksei Hauryliuk, Provat K. Saha, Allen L. Robinson, Albert A. Presto, and R. Subramanian. 2019. 'Fine particle mass monitoring with low-cost sensors: Corrections and long-term performance evaluation', Aerosol Science and Technology: 1-15.
    2. Mehadi, Ahmed, Hans Moosmüller, David E. Campbell, Walter Ham, Donald Schweizer, Leland Tarnay, and Julie Hunter. 2019. 'Laboratory and field evaluation of real-time and near real-time PM2.5 smoke monitors', Journal of the Air & Waste Management Association: 1-22.
    3. Feenstra, Brandon, Vasileios Papapostolou, Sina Hasheminassab, Hang Zhang, Berj Der Boghossian, David Cocker, and Andrea Polidori. 2019. 'Performance evaluation of twelve low-cost PM2.5 sensors at an ambient air monitoring site', Atmospheric Environment, 216: 116946.
  4. In addition, I don’t think there was any discussion of linearity of the PurpleAir-II data or any scatter plots as were shown for the LBNL tests. It would probably be helpful to include at least scatter plots of the data if not linearity metrics as some previous work in smoke impacted areas has shown nonlinear response of Plantower sensors.
    1. Stampfer, Orly, Elena Austin, Terry Ganuelas, Tremain Fiander, Edmund Seto, and Catherine Karr. 2020. 'Use of low-cost PM monitors and a multi-wavelength aethalometer to characterize PM2.5 in the Yakama Nation reservation', Atmospheric Environment: 117292.

  1. It would be helpful to also include a range of concentration measurements in table 1 or somewhere in the paper or this may be covered if you include a scatter plot of the full range.
  2. It looks like the authors forgot to fill in the Author Contributions section
  3. Lines 271-279: Can you quantify any more how you identified problematic sensors?

Minor comments:

  1. It looks like micro in micro grams/meters isn't showing not sure if it's just an issue with my pdf viewer but something to check
  2. PurpleAir is 1 word
  3. Figure 1 caption: Green is misspelled
  4. Line 145: It is approved as a US FEM for 24-hr measurements
  5. Lines 233-234: It appears both links go to the same place. This may be because PurpleAir has discontinued the sensor list?
  6. Figure 8 & 9: It might be helpful if you specified the averaging interval on the error. 1-hr?
  7. Line 524: I think there is a typo in this sentence.

Author Response

Thank you for the positive feedback and many helpful comments. We particularly appreciate the reference citations that were provided. All were on point and informative and have been cited in the revised manuscript.

Responses to detailed comments are provided below.

Comments:

  1. Plantower sensors report PM2.5 with two corrections cf_1 and cf_atm. There is much confusion surrounding these outputs for PurpleAir since the labels were switched until late 2019. This is described in Tryner et al. Can you please specify which cf you are using and whether you are using the current naming convention or the pre 2020 naming convention? This is likely something to define for the Plantower based sensors other than the PurpleAir as well.   
    1. Tryner, Jessica, Christian L'Orange, John Mehaffy, Daniel Miller-Lionberg, Josephine C. Hofstetter, Ander Wilson, and John Volckens. 2020. 'Laboratory evaluation of low-cost PurpleAir PM monitors and in-field correction using co-located portable filter samplers', Atmospheric Environment, 220: 117067.

Response: For the PA-II and PAI sensors we used the CF_1 data stream reported by the Plantower sensor and now reported accurately by PA-II but at the time of the fire it was switched with cf_atm. as cf_1. We confirmed this  For the other monitors that used the Plantower sensors we used the instrument reported PM2.5. And the citation has been added. 

2. It would likely be helpful if you added a table summarizing instruments, acronyms, type (FEM/filter/professional/low-cost etc) and timelines. This may help readers quickly reference monitor acronyms as they come up.

Response: We added Table 1, which provides summary descriptions of each instrument used in the study and removed the acronyms PEM and OPS, which were each used only once following the first use and could be avoided the second time.

3. It seems like a lot of previous work with PurpleAir and other sensors has adjusted the data using linear regression including an intercept significantly different than zero (e.g. Mehadi, Maling, Feenstra). If you have an intercept significantly different than zero it would impact your AF differently at low and high concentration. Did you consider this when doing this analysis?

    1. Malings, Carl, Rebecca Tanzer, Aliaksei Hauryliuk, Provat K. Saha, Allen L. Robinson, Albert A. Presto, and R. Subramanian. 2019. 'Fine particle mass monitoring with low-cost sensors: Corrections and long-term performance evaluation', Aerosol Science and Technology: 1-15.
    2. Mehadi, Ahmed, Hans Moosmüller, David E. Campbell, Walter Ham, Donald Schweizer, Leland Tarnay, and Julie Hunter. 2019. 'Laboratory and field evaluation of real-time and near real-time PM2.5 smoke monitors', Journal of the Air & Waste Management Association: 1-22.
    3. Feenstra, Brandon, Vasileios Papapostolou, Sina Hasheminassab, Hang Zhang, Berj Der Boghossian, David Cocker, and Andrea Polidori. 2019. 'Performance evaluation of twelve low-cost PM2.5 sensors at an ambient air monitoring site', Atmospheric Environment, 216: 116946.

Response: We agree with the reviewer that this point should be addressed. We introduce the issue of zero vs. non-zero intercepts when presenting results from measurements with several low-cost and two professional grade monitors at Lawrence Berkeley National Lab. On p.10 of the revision we present fits to LBNL data using zero or non-zero intercepts. This point is reinforced with a restructuring of the former Fig 3, which has been moved so Supporting Info as Fig S6. (Removed to accommodate the addition of Table 1 requested in prior comment.) Results in the figure moved to SI are similar to the former Fig 4, now Fig 3. We also restructured the former Table 1 (now Table 2) to include linear fits with zero and non-zero intercepts to relate PA-II data to reference measurements. The slopes of these fits provide slightly different adjustment factors than the medians of adjustment factors calculated from ratios of measurements over 4-h intervals over the course of the fire. The figure was removed from the paper because the relationships conveyed in that figure are also conveyed in Figure 4. The references provided by the reviewer have been added to the discussion on p. 10.   

4. In addition, I don’t think there was any discussion of linearity of the PurpleAir-II data or any scatter plots as were shown for the LBNL tests. It would probably be helpful to include at least scatter plots of the data if not linearity metrics as some previous work in smoke impacted areas has shown nonlinear response of Plantower sensors.

    1. Stampfer, Orly, Elena Austin, Terry Ganuelas, Tremain Fiander, Edmund Seto, and Catherine Karr. 2020. 'Use of low-cost PM monitors and a multi-wavelength aethalometer to characterize PM2.5 in the Yakama Nation reservation', Atmospheric Environment: 117292.

Response: Thanks for calling attention to this. In fact, we did see a non-linear response of the PA-II monitors at the highest smoke levels. We added FIgure S9 to the SI to show these data for all sites. Since the error of the simple linear adjustment predominantly impacts estimates at PM2.5 concentrations above roughly 180 ug/m3, which is solidly in the unhealthy realm. The added complexity is not needed. This is noted in the text at the bottom of p. 13..  

5. It would be helpful to also include a range of concentration measurements in table 1 or somewhere in the paper or this may be covered if you include a scatter plot of the full range.

Response: We added the 10th-90th range in Table 2 and added Figure S9 to the SI which shows the distributions of data across the sites. Figure 6 also shows concentrations. 

6. It looks like the authors forgot to fill in the Author Contributions section

Response: This has been added.

7. Lines 271-279: Can you quantify any more how you identified problematic sensors?

Response: Visual inspection with a confirmation of poor correlations by calculating the R2. R2 below 0.6 confirmed a poor sensor.

Minor comments:

8. It looks like micro in micro grams/meters isn't showing not sure if it's just an issue with my pdf viewer but something to check

Response: This appears to be a conversion issue as the symbol shows correctly in the submitted MS Word document.

9. PurpleAir is 1 word

Response: this has been corrected throughout.

10. Figure 1 caption: Green is misspelled

Response: Thank you for catching that; it has been fixed.

11. Line 145: It is approved as a US FEM for 24-hr measurements

Response: This clarification has been added.

12. Lines 233-234: It appears both links go to the same place. This may be because PurpleAir has discontinued the sensor list?

Response: Thank you for calling our attention to this. It appears that PurpleAir has changed their pathway for accessing data. Since individual sensor data are now accessed via the map, we removed the second URL from the text. 

13. Figure 8 & 9: It might be helpful if you specified the averaging interval on the error. 1-hr?

Response: We clarified in the captions that the figures show distributions of errors in 4h average concentrations resulting from no adjustment or using the specific adjustment factors. 

14. Line 524: I think there is a typo in this sentence.

Response: I don’t see a typo on line 524. Perhaps there has been a shift in the line numbers…?

Reviewer 3 Report

The manuscript entitled “Wildfire Smoke Adjustment Factors for Low-Cost and Professional PM2.5 Monitors with Optical Sensors”, has developed adjustments factors for adjusting the concentrations of PM2.5 measured by different grades of sensors. Generally, it is a timely and needed research. However, several aspects are not clear (poorly illustrated), which need addressing before it can be published.

  1. The link provided (https://www.epa.gov/naaqs/particulate-matter-pm-air-quality-standards) is not correct, the correct link is: https://www.epa.gov/criteria-air-pollutants/naaqs-table. Also, the citation is not correct according to the journal referencing/citation rules.
  2. The standard provided 12g/m3 and 35 g/m3 are not correct. These should be in micro-gram/m3 (ug/m3). I think this is probably caused by the PDF conversion process. Check the whole paper for such errors.
  3. Line 126: “impacted air quality monitoring stations”, what does this mean? Does this refer to low-cost sensors? Or to the areas impacted by fire smoke.
  4. What is LBNL? It should be defined at first appearance.
  5. Line 154: Federal Equivalent Method – once an acronym/abbreviation is defined at its first appearance, then the authors should stick to the abbreviation. Check the whole paper for such examples. Too many abbreviations have been used, which make the paper confusing and hard to understand. Several abbreviations have been defined but never used in later text. You need to define abbreviation only for those terms which are frequently used in the paper.
  6. Line 202-203: “During three periods multi-hour periods totaling 26 h, the two doors were opened to increase indoor PM2.5 to levels close to those measured outdoors at nearby regulatory monitoring stations”. In my opinion indoor and outdoor levels of pollutants will never be equal, even if you open the door.
  7. In methodology section, the authors have not described the statistical approaches used for the analysis and creating adjustment factors.
  8. Each Figure should be able to stand alone. The details provided in the captions should explain the abbreviations etc. (e.g., Figure 2, here each abbreviation should be defined).
  9. The serious weakness of the paper is that sensors installed at different type of environment are compared and used for developing adjustment factors. On top, indoor and outdoor sensors are compared. In my opinion only collocated sensors can be used for developing adjustment factors, otherwise the adjustment factors would be invalid.
  10. The authors talk about only wildfire smoke, how they adjusted for other emissions?
  11. “””Author Contributions: For research articles with several authors, a short paragraph specifying their individual contributions must be provided. The following statements should be used “conceptualization, X.X. and Y.Y.; methodology, X.X.; software, X.X.; validation, X.X., Y.Y. and Z.Z.; formal analysis, X.X.; investigation, X.X.; resources, X.X.; data curation, X.X.; writing—original draft preparation, X.X.; writing—review and editing, X.X.; visualization, X.X.; supervision, X.X.; project administration, X.X.; funding acquisition, Y.Y.”, please turn to the CRediT taxonomy for the term explanation. Authorship must be limited to those who have contributed substantially to the work reported”””,

what is this????????????? Kindly replace text by your own text, defining the contribution of each author to the paper.

  1. Check the reference style. Also in several places in the text website links are inserted. They should be moved to the reference list. In the text only a number should be inserted as the other references.

Author Response

The manuscript entitled “Wildfire Smoke Adjustment Factors for Low-Cost and Professional PM2.5 Monitors with Optical Sensors”, has developed adjustments factors for adjusting the concentrations of PM2.5 measured by different grades of sensors. Generally, it is a timely and needed research. However, several aspects are not clear (poorly illustrated), which need addressing before it can be published.

Thank you for your detailed comments, which helped us to improve the paper.

1. The link provided (https://www.epa.gov/naaqs/particulate-matter-pm-air-quality-standards) is not correct, the correct link is: https://www.epa.gov/criteria-air-pollutants/naaqs-table. Also, the citation is not correct according to the journal referencing/citation rules.

Response: These have been replaced with numbered citations

2. The standard provided 12g/m3 and 35 g/m3 are not correct. These should be in micro-gram/m3 (ug/m3). I think this is probably caused by the PDF conversion process. Check the whole paper for such errors.

Response: This does appear to be a pdf conversion issue. We tried to identify and fix all such errors. 

3. Line 126: “impacted air quality monitoring stations”, what does this mean? Does this refer to low-cost sensors? Or to the areas impacted by fire smoke.

Response: The word “impacted” was removed.

4. What is LBNL? It should be defined at first appearance.

Response: This has been clarified.

5. Line 154: Federal Equivalent Method – once an acronym/abbreviation is defined at its first appearance, then the authors should stick to the abbreviation. Check the whole paper for such examples. Too many abbreviations have been used, which make the paper confusing and hard to understand. Several abbreviations have been defined but never used in later text. You need to define abbreviation only for those terms which are frequently used in the paper.

Response: We removed this second definition of FEM. We removed PEM and OPS acronyms.

6. Line 202-203: “During three periods multi-hour periods totaling 26 h, the two doors were opened to increase indoor PM2.5 to levels close to those measured outdoors at nearby regulatory monitoring stations”. In my opinion indoor and outdoor levels of pollutants will never be equal, even if you open the door.

Response: text changed to read “to be closer to outdoors”.

7. In methodology section, the authors have not described the statistical approaches used for the analysis and creating adjustment factors.

Response: The analysis used simple statistical approaches and common summary measures such as means, medians, percentiles, errors, etc. For efficiency, the metrics used are generally noted as the results are presented. 

8. Each Figure should be able to stand alone. The details provided in the captions should explain the abbreviations etc. (e.g., Figure 2, here each abbreviation should be defined).

Response: The following was added to this caption: “Refer to Table 1 for monitor descriptions”

9. The serious weakness of the paper is that sensors installed at different type of environment are compared and used for developing adjustment factors. On top, indoor and outdoor sensors are compared. In my opinion only collocated sensors can be used for developing adjustment factors, otherwise the adjustment factors would be invalid.

Response: The monitors at LBNL were collocated. PA-II monitors were at varying distances from their paired AQS stations and included some - in Sacramento - there were collocated. 

10. The authors talk about only wildfire smoke, how they adjusted for other emissions?

Response: The analysis focused on periods when the ambient PM2.5 was overwhelmingly comprised of wildfire smoke. Our group has published separately on the response of low-cost and professional grade IAQ monitors for residential sources of PM. (Wang et al 2020)  

11. “””Author Contributions: For research articles with several authors, a short paragraph specifying their individual contributions must be provided. The following statements should be used “conceptualization, X.X. and Y.Y.; methodology, X.X.; software, X.X.; validation, X.X., Y.Y. and Z.Z.; formal analysis, X.X.; investigation, X.X.; resources, X.X.; data curation, X.X.; writing—original draft preparation, X.X.; writing—review and editing, X.X.; visualization, X.X.; supervision, X.X.; project administration, X.X.; funding acquisition, Y.Y.”, please turn to the CRediT taxonomy for the term explanation. Authorship must be limited to those who have contributed substantially to the work reported”””,

what is this????????????? Kindly replace text by your own text, defining the contribution of each author to the paper. 

Response: This section has been added.

12. Check the reference style. Also in several places in the text website links are inserted. They should be moved to the reference list. In the text only a number should be inserted as the other references.

Response: these have been changed to citations.

Round 2

Reviewer 3 Report

I am happy with the revised version of the manuscript.